# Identifiability of Sparse Causal Effects using Instrumental Variables

**Niklas Pfister**[1, *]                    **Jonas Peters**[1, *]

[1]Department of Mathematical Sciences, University of Copenhagen, Denmark
[*]Authors contributed equally.

## Abstract

Exogenous heterogeneity, for example, in the form of instrumental variables can help us learn a system's underlying causal structure and predict the outcome of unseen intervention experiments. In this paper, we consider linear models in which the causal effect from covariates $X$ on a response $Y$ is sparse. We provide conditions under which the causal coefficient becomes identifiable from the observed distribution. These conditions can be satisfied even if the number of instruments is as small as the number of causal parents. We also develop graphical criteria under which identifiability holds with probability one if the edge coefficients are sampled randomly from a distribution that is absolutely continuous with respect to Lebesgue measure and $Y$ is childless. As an estimator, we propose `spaceIV` and prove that it consistently estimates the causal effect if the model is identifiable and evaluate its performance on simulated data. If identifiability does not hold, we show that it may still be possible to recover a subset of the causal parents.

## 1 INTRODUCTION

Instrumental variables [Wright, 1928, Imbens and Angrist, 1994, Newey, 2013] allow us to consistently estimate causal effects from covariates $X$ on a response $Y$ even if the covariates and response are connected through hidden confounding. These approaches usually rely on identifying moment equations such as $\mathrm{Cov}[I, Y - X^\top \beta] = 0$ with $I$ being the instrumental variable (IV). Under some assumptions such as the exclusion restriction, this equation is satisfied for the true causal coefficient $\beta = \beta^*$; in a linear setting, for example, this is the case if we can write $Y = X^\top \beta^* + g(H, \varepsilon^Y)$ with $H, \varepsilon^Y$ being independent of $X$ and $I$ and $\varepsilon^Y$ independent of

$X$. Identifiability of $\beta^*$, however, requires that the moment equation is not satisfied for any other $\beta \neq \beta^*$. Formally, this condition is often written as a rank condition on the covariance between $I$ and $X$, which implies that the dimension of $I$ must be at least as large as the number of components of $X$.

In this work, we consider the case where the causal coefficient $\beta^*$ is assumed to be sparse. This assumption allows us to relax existing identifiability conditions: it is, for example, possible to identify $\beta^*$ even if there are much less instruments than covariates. Our results are proved in the context of linear structural causal models (SCMs) [Pearl, 2009, Bongers et al., 2021], that is, we also assume linearity among the $X$ variables. We prove sufficient conditions for identifiability of $\beta^*$ that are based on rank conditions of the matrix of causal effects from $I$ on the parents of $Y$. We then investigate for which graphical structures we can expect such conditions to hold. Consider, for example, the graph shown in Figure 1. Square nodes represent instruments, and hidden variables between variables in $X \cup \{Y\}$ can exist but are not drawn (we formally introduce such graphs in Section 2.1). Sparse identifiability in this graph is not obvious: Is the causal effect from the parents of $Y$ to $Y$ generically identifiable if the true underlying and unknown graph is the one shown (including the two dashed edges)? And what about the graph excluding the two dashed edges? We translate the rank conditions for identifiability to structural SCMs whose coefficients are drawn randomly from a distribution that is absolutely continuous with respect to Lebesgue measure. This allows us to develop graphical criteria that can answer these questions.

If identifiability holds, the causal effect can be estimated from data. We propose an estimator called `spaceIV` ('**spa**rse **c**ausal **e**ffect **IV**'). It is based on the limited information maximum likelihood (LIML) estimator [Anderson and Rubin, 1949, Amemiya, 1985]. This estimator has similar properties as the two stage least squares estimator and has the same asymptotic normal distribution, for example [Mariano, 2001]. But as it minimizes the Anderson-Rubin

*Accepted for the 38th Conference on Uncertainty in Artificial Intelligence* (UAI 2022).

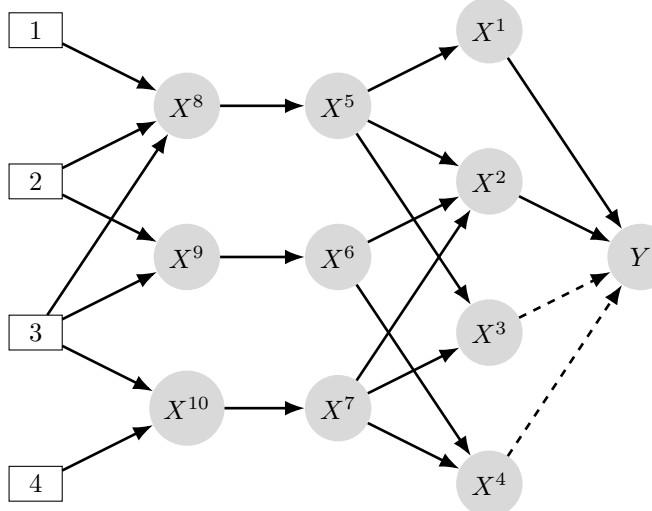

Figure 1: Graphical representation of two linear SCMs, as described in Section 2.1 (hidden variables between $X$ and $Y$ variables exist but are not drawn). If the data come from a system corresponding to the unknown graph including (or excluding) dashed edges, can we identify the causal effect from $X$ to $Y$ from the joint distribution over $I$, $X$, and $Y$? These questions are discussed in Example 4.

## 2 IV MODELS WITH SHIFT INTERVENTIONS

Consider the following structural causal model (SCM)

$$
\begin{aligned}
X &:= BX + AI + h(H, \varepsilon^X) \\
Y &:= X^\top \beta^* + g(H, \varepsilon^Y),
\end{aligned}
\tag{1}
$$

where $h$ and $g$ are arbitrary measurable functions and $\mathrm{Id} - B$[1] is invertible. Here, $X \in \mathbb{R}^d$ denotes the observed variables, $H \in \mathbb{R}^q$ the unobserved variables, $I \in \mathbb{R}^m$ the instrumental variables (following an $m$-dimensional distribution, which is not modelled explicitly), $Y \in \mathbb{R}$ the response and $I$, $H$, $\varepsilon^X$ and $\varepsilon^Y$ are jointly independent and assume that the covariates are non-descendants of $Y$ (see also Remark 8). In contrast to classical IV settings, we thus explicitly model the causal effects of the instruments $I$ on the predictor variables $X$. Throughout the paper, we assume that $\mathrm{Cov}[I]$ is invertible. We assume that we have access to an i.i.d. data set $(X_1, Y_1, I_1), \ldots, (X_n, Y_n, I_n)$ sampled from the induced distribution and are interested in estimating the causal effect $\beta^*$. We call the set of non-zero components of $\beta^*$ the parents of $Y$ and denote it by $\mathrm{PA}(Y)$.

Our model covers the case, where we observe data from $m$ different experiments, each of which corresponds to a fixed intervention shift. More precisely, we can choose $I$ such that for all $k \in \{1, \ldots, m\}$, we have $P(I = e_k) = 1/m$, with $e_k$, $k \in \{1, \ldots, m\}$, being the $k$-th unit vector in $\mathbb{R}^m$. Here, each column in the matrix $A$ specifies a different experiment in which (a subset of) the $X$ variables is shifted by the amount specified in that column.

### 2.1 GRAPHICAL REPRESENTATION

Given a data generating process of the form (1), we represent it graphically as follows: Each of the $d$ components[2] of $X$ is represented by a node, which we call a *prediction node*. There is a directed edge from $X^i$ to $X^j$ if and only if $B_{ji} \neq 0$. In addition, we represent the $k$th component of $I$ by a square node with label '$k$', which we call *instrument node*. There is a directed edge from $k$ to $X^j$ if and only if $A_{j,k} \neq 0$. (There are no connections between instrument nodes, even though they may be dependent.) Finally, we represent the response $Y$ with the same node style as is used for the predictors and include a directed edge from $X^j$ to $Y$ if and only if $\beta^j \neq 0$. In the graph, we do not represent hidden variables (even though they are allowed to exist). Consequently, such graphs do not satisfy the Markov condition [e.g., Lauritzen, 1996].

test statistic, it allows us to prove theoretical guarantees. We evaluate the performance of spaceIV on simulated data. If identifiability does not hold, we prove that it may still be possible to identify a subset of the causal parents of $Y$.

Numerous extensions to the classical linear instrumental variable setting have been proposed. For example, nonlinear effects [Imbens and Newey, 2009, Dunker et al., 2014, Torgovitsky, 2015, Loh, 2019, Christiansen et al., 2020] have been considered, often in relation with higher order moment equations [Hartford et al., 2017, Singh et al., 2019, Bennett et al., 2019, Muandet et al., 2020, Saengkyongam et al., 2022]. Furthermore, Belloni et al. [2012], Mckeigue et al. [2010] assume that the effect from the instruments on the covariates is sparse. For example, it has been shown that consistent estimators exist if at least half of all instruments are valid [Kang et al., 2016]. To the best of our knowledge, while existing work considers sparsity constraints between the instruments and the covariates ('first stage'), the assumption of a sparse causal effect ('second stage') and its benefits has not yet been analyzed.

Our paper is structured as follows. Section 2 introduces the formal setup. Section 3.1 presents the main identifiability result for sparse causal effect models and Section 4 develops the corresponding graphical criteria. Section 5 introduces the estimator spaceIV and Section 6 includes simulation experiments. Code is attached as supplementary material.

---

[1]Here Id denotes the identity matrix.

[2]In a slight abuse of notation, we sometimes identify each component with its index.

*Accepted for the 38th Conference on Uncertainty in Artificial Intelligence* (UAI 2022).

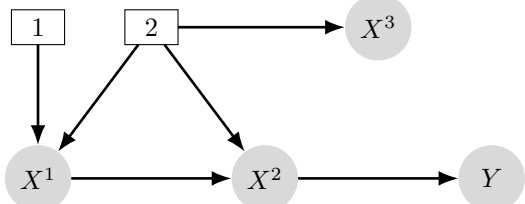

Figure 2: Graphical representation of Example 1, as described in Section 2.1 (the hidden variable $H$ is omitted).

**Example 1.** *Consider an SCM of the following form*

$$\begin{pmatrix} X^1 \\ X^2 \\ X^3 \end{pmatrix} := \begin{pmatrix} 0 \\ b_{21}X^1 \\ 0 \end{pmatrix} + \begin{pmatrix} a_{11} & a_{12} \\ 0 & a_{22} \\ 0 & a_{32} \end{pmatrix} \begin{pmatrix} I^1 \\ I^2 \end{pmatrix} + h(H, \varepsilon^X)$$

$$Y := \begin{pmatrix} X^1 & X^2 & X^3 \end{pmatrix} \begin{pmatrix} 0 \\ \beta_2^* \\ 0 \end{pmatrix} + g(H, \varepsilon^Y), \qquad (2)$$

*where $I^1$, $I^2$, $H$, $\varepsilon^Y$, $\varepsilon^X$ are jointly independent. Figure 2 shows the corresponding graphical representation.*

# 3 IDENTIFIABILITY IN SPARSE-EFFECT IV MODELS

Consider a data generating process of the form (1). Because the intervention $I$ does not directly enter the structural assignment of $Y$ and $(H, \varepsilon^Y, I)$ are jointly independent, the causal coefficient $\beta^*$ satisfies the *moment condition*

$$\operatorname{Cov}\left(I, Y - X^\top \beta^*\right) = 0. \qquad (3)$$

The solution space of the moment condition is given by

$$\mathcal{B} := \{\beta \in \mathbb{R}^d \mid \operatorname{Cov}(I, X)\beta = \operatorname{Cov}(I, Y)\}.$$

It can be shown that this is a $(d - \operatorname{Rank}(A))$-dimensional space. The true causal coefficient $\beta^*$ is therefore identified by (3) if and only if $\operatorname{Rank}(A) = d$. This directly implies that the number of instruments needs to be greater or equal to the number of predictors, a well-known necessary condition for identifiability in the linear IV model.

In this work, we investigate the case where $\mathcal{B}$ is allowed to be non-degenerate. To analyse conditions for identifiability, we define the $(m \times d)$-matrix

$$C := A^\top (\operatorname{Id} - B)^{-\top}. \qquad (4)$$

The entry $C_{i,j}$ corresponds to the the $i$-th component of the total causal effect from $I$ onto $X^j$ in the SCM given in (1). This entry correspond to summing over all directed paths from instrument node $i$ to $X^j$ and for each path, multiplying the coefficients. The matrix $C$ will play a central role when analyzing identifiability. For example, using the matrix $C$,

Proposition 2 characterizes settings under which individual components of the causal coefficient $\beta^*$ are identifiable. This result does not require any additional assumptions on the underlying model. In Section 3.1, we then show that if the causal coefficient $\beta^*$ is sparse (i.e., it contains many zeros) it can still be identifiable even if $\mathcal{B}$ is non-degenerate.

**Proposition 2** (Partial identifiability of causal coefficient). *Consider a data generating process of the form* (1). *Then, for all $j \in \{1, \ldots, d\}$ it holds that*

$$\beta_j^* \text{ is identifiable by (3)} \quad \Leftrightarrow \quad \operatorname{Null}(C)_j = \{0\},$$

*where $\operatorname{Null}(C)_j$ denotes the $j$-th coordinate of the null space of $C$. Moreover, whenever $\operatorname{Null}(C)_j = \{0\}$ it holds that $\beta_j^* = (\operatorname{Cov}(I, X)^\dagger \operatorname{Cov}(I, Y))_j$, where $(\cdot)^\dagger$ denotes the Moore-Penrose inverse.*

The proof can be found in Appendix A.

## 3.1 IDENTIFIABILITY OF SPARSE CAUSAL COEFFICIENTS

We have argued that the causal parameter is in general not fully identified by the moment condition (3). However, we can obtain identifiability by additionally assuming that the causal coefficient $\beta^*$ is sparse. To make this more precise, consider the following optimization

$$\min_{\beta \in \mathcal{B}} \|\beta\|_0. \qquad (5)$$

As we will see below, under mild conditions on the interventions $I$, the causal coefficient $\beta^*$ is a unique solution to this problem.

We now make the following assumptions[3].

(A1) It holds that $\operatorname{Rank}\left(C_{\operatorname{PA}(Y)}\right) = |\operatorname{PA}(Y)|$.

(A2) For all $S \subseteq \{1, \ldots, d\}$ it holds that

$$\left.\begin{array}{l} \operatorname{Rank}(C_S) \leq \operatorname{Rank}\left(C_{\operatorname{PA}(Y)}\right) \text{ and} \\ \operatorname{Im}(C_S) \neq \operatorname{Im}\left(C_{\operatorname{PA}(Y)}\right) \end{array}\right\} \text{ implies}$$
$$\left\{\forall w \in \mathbb{R}^{|S|}: \quad C_S w \neq C_{\operatorname{PA}(Y)} \beta_{\operatorname{PA}(Y)}^* \right. .$$

(A3) For all $S \subseteq \{1, \ldots, d\}$ with $|S| = |\operatorname{PA}(Y)|$ and $S \neq \operatorname{PA}(Y)$ we have $\operatorname{Im}(C_S) \neq \operatorname{Im}\left(C_{\operatorname{PA}(Y)}\right)$.

(A1) is a necessary assumption in order to identify $\beta^*$; it guarantees that an IV regression based on the true parent set $\operatorname{PA}(Y)$ identifies the correct coefficients. (A2) is an assumption on the underlying causal model that ensures

---

[3]Here we use the convention that for a matrix $D \in \mathbb{R}^{m \times d}$ and a subset $S \subseteq \{1, \ldots, d\}$ the subindexed matrix $D_S$ corresponds to the $m \times |S|$-submatrix of $D$ consisting of all columns that are indexed by $S$ and $\operatorname{Im}(D)$ denotes the image of $D$.

*Accepted for the 38th Conference on Uncertainty in Artificial Intelligence* (UAI 2022).

that certain types of cancellation cannot occur. It is a rather mild assumption in the following sense: if one considers the true causal parameter $\beta^*$ as randomly drawn from a distribution absolutely continuous with respect to Lebesgue measure it would almost surely lead to a system that satisfies (A2) (see Proposition 9 in Appendix B). As shown in the following theorem, (A1) and (A2) are sufficient to ensure that $\beta^*$ solves (5). Additionally assuming (A3) ensures that the solution is unique; it can be seen as requiring an extra level of heterogeneity in how the interventions affect the system (see also Section 4).

**Theorem 3** (Identifiability of sparse causal parameters). *Consider a data generating process of the form* (1). *If (A1) and (A2) hold, then $\beta^*$ is a solution to* (5). *Moreover, if, in addition, (A3) holds, then $\beta^*$ is the unique solution.*

*Proof.* We use the notation $\xi^X := h(H, \varepsilon^X)$ and $\xi^Y := g(H, \varepsilon^X)$. Then (1) and the assumption of joint independence of $I$, $\xi^X$, and $\xi^Y$ imply that

$$\text{Cov}[I, X] = \text{Cov}\left[I, (\text{Id} - B)^{-1}(AI + \xi^X)\right]$$
$$= \text{Cov}[I]A^\top(\text{Id} - B)^{-\top}. \qquad (6)$$

Similarly, we get that

$$\text{Cov}[I, Y] = \text{Cov}\left[I, (AI + \xi^X)^\top(\text{Id} - B)^{-\top}\beta^* + \xi_Y\right]$$
$$= \text{Cov}\left[I, \beta^{*\top}(\text{Id} - B)^{-1}(AI + \xi^X) + \xi_Y\right]$$
$$= \text{Cov}[I]A^\top(\text{Id} - B)^{-\top}\beta^*. \qquad (7)$$

Hence, for any $\tilde{\beta} \in \mathcal{B}$, using the definition of $\mathcal{B}$ and combining (6) and (7) we get that

$$\text{Cov}[I]C\tilde{\beta} = \text{Cov}[I]C\beta^*.$$

Here, we used the definition of $C$ in (4). As $\text{Cov}[I]$ is invertible, we get

$$C\tilde{\beta} = C\beta^*, \qquad (8)$$

Furthermore, it holds for[4] $S = \text{supp}(\tilde{\beta})$ that

$$C_S\tilde{\beta}_S = C_{\text{PA}(Y)}\beta^*_{\text{PA}(Y)}. \qquad (9)$$

We now prove the first part of the theorem. Assume (A1) and (A2) are satisfied. We want to show that

$$\beta^* \in \underset{\beta \in \mathcal{B}}{\arg\min}\|\beta\|_0. \qquad (10)$$

Since $\beta^* \in \mathcal{B}$, it is sufficient to show that for all $\tilde{\beta} \in \mathcal{B}$ it holds that $\|\tilde{\beta}\|_0 \geq |\text{PA}(Y)|$. To this end, fix $\tilde{\beta} \in \mathcal{B}$ and set $S = \text{supp}(\tilde{\beta})$. For the sake of contradiction assume $|S| < |\text{PA}(Y)|$, then using (A1) we get that

$$\text{Rank}\left(C_{\text{PA}(Y)}\right) = \dim(\text{Im}\left(C_{\text{PA}(Y)}\right)) = |\text{PA}(Y)|$$
$$> S \geq \dim(\text{Im}(C_S)) = \text{Rank}(C_S).$$

---

[4]The support of a vector is defined as the set of indices of non-zero elements.

This implies $\text{Rank}\left(C_{\text{PA}(Y)}\right) \geq \text{Rank}(C_S)$ and $\text{Im}\left(C_{\text{PA}(Y)}\right) \neq \text{Im}(C_S)$. Thus, by (A2), this contradicts (9). This completes the first part of the proof.

Next, we prove the second part of the theorem. Assume that (A1), (A2) and (A3) are satisfied. By the previous part of the proof, we have seen that $\beta^*$ satisfies (10). It therefore only remains to show that there is no other solution. Assume for the sake of contradiction that there exists $\tilde{\beta} \in \mathcal{B}$ with $S := \text{supp}(\tilde{\beta})$ such that $|S| = |\text{PA}(Y)|$ and $S \neq \text{PA}(Y)$. Then by (A3) we have $\text{Im}(C_S) \neq \text{Im}\left(C_{\text{PA}(Y)}\right)$. By (A1) it holds that

$$\text{Rank}\left(C_{\text{PA}(Y)}\right) = |\text{PA}(Y)| = |S| \geq \text{Rank}(C_S).$$

Hence, together with the condition $\text{Im}(C_S) \neq \text{Im}\left(C_{\text{PA}(Y)}\right)$ we can use (A2) to get a contradiction to (9). This completes the proof of Theorem 3. $\square$

Section 5.1 shows how one can identify a subset of the causal parents under even milder conditions. Remark 8 discusses the case where the covariates can also be descendants of $Y$.

## 4 GRAPHICAL CHARACTERIZATION

We now formulate the identifiability result from Section 3.1 in graphical terms. Suppose we are given a data generating process of the form (1) with corresponding graph $\mathcal{G}$ (as described in Section 2.1), which in this section is assumed to be acyclic. The parents of $Y$ are denoted by $\text{PA}(Y)$ and correspond to the non-zero entries of $\beta^*$. Moreover, for any set $S \subseteq \{1, \ldots, d\}$, we define the set of all *intervention ancestors* of variables in $S$ as

$$\text{AN}_I[S] := \{j \in \{1, \ldots, m\} \mid j \in \text{AN}(S)\}.[5]$$

This set contains the instrument nodes that are ancestors of $S$.

We can now state the following graphical assumptions.

(B1) There are at least $|\text{PA}(Y)|$ disjoint directed paths (not sharing any node) from $I$ to $\text{PA}(Y)$.

(B2) The non-zero coefficients of the causal coefficient $\beta^*_{\text{PA}(Y)} \in \mathbb{R}^{|\text{PA}(Y)|}$ and the non-zero entries of $A$ and $B$ are randomly drawn from a distribution $\mu$ which is absolutely continuous with respect to Lebesgue measure (and are independent of the other variables).

(B3) For all $S \subseteq \{1, \ldots, d\}$ with $|S| = |\text{PA}(Y)|$ and $S \neq \text{PA}(Y)$ at least one of the following conditions is satisfied

---

[5]$\text{AN}(S)$ denotes the ancestor set of $S$ consisting of all nodes with a directed path to a node in $S$. Throughout the paper, we use the convention that a node is contained in the set of its ancestors.

*Accepted for the 38th Conference on Uncertainty in Artificial Intelligence* (UAI 2022).

(i) $\mathrm{AN}_I[S] \neq \mathrm{AN}_I[\mathrm{PA}(Y)]$.

(ii) The smallest set $T$ of nodes such that all directed paths from $I$ to $\mathrm{PA}(Y)$ and from $I$ to $S$ go through $T$ is of size at least $|\mathrm{PA}(Y)| + 1$.

We will see in Theorem 5 below that the causal effect becomes identifiable if (B1)–(B3) hold. Let us discuss these assumptions using a few examples.

**Example 4.** *(i) The example from Example 1 and Figure 2 is discussed in Figure 7 in Appendix E.*

*(ii) Figure 3 contains another identifiable example.*

*(iii) We now come back to the example graphs shown in Figure 1. Consider an SCM with the graph structure shown including the dashed edges. (B1) is violated, as the effect of the four instruments is 'channelled' through three variables. Indeed, here, the causal effect from $(X^1, X^2, X^3, X^4)$ on $Y$ is in general non-identifiable – even though all instruments are connected to all causal parents of $Y$ (the rank of $C_{\mathrm{PA}(Y)}$ is three and therefore too small to identify $\beta^*$).*

*(iv) Consider an SCM with the graph structure shown in Figure 1 (dashed edges not included). Here, (B1) holds. (B3) is satisfied, too: e.g., for the set $S := \{X^3, X^4\}$, we have $\mathrm{AN}_I[\{X^3, X^4\}] = \mathrm{AN}_I[\{X^1, X^2\}]$, so (B3) (i) is violated, but (B3) (ii) holds (which implies $\mathrm{Im}(C_S) \neq \mathrm{Im}(C_{\mathrm{PA}(Y)})$, see proof of Theorem 5): there is no set of size two such that all directed paths go through this set (note that $|\mathrm{AN}(\{X^3, X^4\})| = 3$, for example). Thus, if additionally (B2) holds, then $\beta^*$ is identifiable.*

A graphical marginalization of graphs similar to the latent projection [Richardson, 2003, Verma, 1993] may help to gain further intuition about the assumptions. Consider a subset $V \subseteq \{1, \ldots, d\}$ of the covariates. The marginalized graph $\mathcal{G}^V$ is then constructed from $\mathcal{G}$ by the following procedure: (i) $\mathcal{G}^V$ consists of all instrument nodes $k$ from $\mathcal{G}$, all predictor nodes $X^j$ from $\mathcal{G}$ for which $j \in V$, and node $Y$; (ii) $\mathcal{G}^V$ contains a directed edge from $X^i$ to $X^j$ if and only if $\mathcal{G}$ contains a directed path from $X^i$ to $X^j$ that does not have any intermediate nodes in $V$ (e.g., because there are no intermediate nodes); (iii) $\mathcal{G}^V$ contains a directed edge from $k$ to $X^j$ if and only if $\mathcal{G}$ contains a directed path from $k$ to $X^j$ that does not have any intermediate nodes in $V$ (e.g., because there are no intermediate nodes). The set $\mathrm{PA}_I^V[U]$ denotes the $\mathcal{G}^V$-parents of $U$ that are intervention nodes: $\mathrm{PA}_I^V[U] := \mathrm{PA}_{\mathcal{G}^V} \cap I$. Figure 7, in Appendix E, shows the marginalized graph corresponding to Example 1 and Figure 2.

(B1) ensures that there is sufficient heterogeneity coming from instruments. In particular, there need to be as many instruments as parents of $Y$ and for all $S \subseteq \mathrm{PA}(Y)$, we have $|\mathrm{PA}_I^{\mathrm{PA}(Y)}[S]| \geq |S|$. In particular, this implies that for all

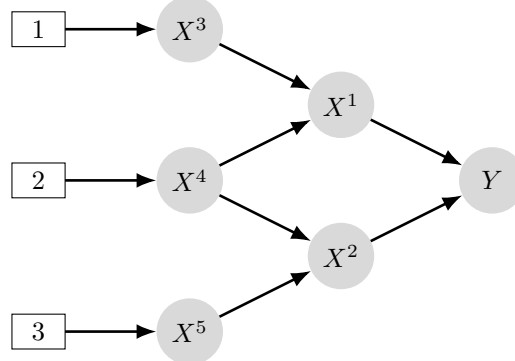

Figure 3: Graphical representation of an example SCM, as described in Section 2.1 (there may be hidden variables between all predictor variables). Here, (B1) holds as there are two distinct paths from $I$ to $\mathrm{PA}(Y)$. (B3) is satisfied, too: $\{X^1, X^5\}$ and $\{X^3, X^2\}$ are the only sets $S$ violating (B3) (i) (because $\mathrm{AN}_I[\{X^1, X^5\}] = \mathrm{AN}_I[\{X^1, X^2\}] = \mathrm{AN}_I[\{X^3, X^2\}]$), but they satisfy (B3) (ii). Thus, given (B2), the causal effect from $(X^1, X^2)$ on $Y$ is identifiable – even though there are less instruments than covariates.

$k \in \mathrm{PA}(Y)$, we have $\mathrm{PA}_I^{\mathrm{PA}(Y)}[k] \neq \emptyset$. In general, however, this is not sufficient for identifiability (see Section 6).

We can now state the graphical version of Theorem 3.

**Theorem 5** (Identifiability of sparse causal coefficients (graph version)). *Consider a data generating process of the form* (1). *If (B1) and (B2) hold, then (A1) and (A2) hold $\mu$-almost surely. Moreover, if, in addition, (B3) holds, then (A3) holds $\mu$-almost surely.*

Together with Theorem 3 this implies that under (B1) and (B2), $\beta^*$ is $\mu$-almost surely a solution to (5) and (B1), (B2), and (B3), it is $\mu$-almost surely the unique solution.

*Proof.* Regarding (A1): With respect to (A1), consider first the marginalization of model (1) over $\mathrm{PA}(Y)$. To do so, we repeatedly substitute $X^j$, $j \in \{1, \ldots, d\}$ with its assignment, that is, the corresponding right-hand side of (1) and obtain

$$X^{\mathrm{PA}(Y)} := C^{\times\top} I + h^\times(H, \varepsilon^X). \tag{11}$$

We then have

$$C_{\cdot,\mathrm{PA}(Y)} = C^\times, \tag{12}$$

where $C_{\cdot,\mathrm{PA}(Y)}$ is the matrix constructed from the columns of $C$ corresponding to $\mathrm{PA}(Y)$. Equality (12) holds by construction: The element of $C_{\cdot,\mathrm{PA}(Y)}$ in row $i$ and the column corresponding to $X^j \in \mathrm{PA}(Y)$ equals the $i$-th component of the total causal effect from $I$ on $X^j$; this is exactly the same in the marginalized model (11). We now argue that $C^\times$ has full rank $\mu$-almost surely. To do so, we perform a more careful replacement scheme that allows us to write

$$X^{\mathrm{PA}(Y)} = C_1 \cdot C_2 \cdot \ldots \cdot C_f \cdot I + h^\times(H, \varepsilon^X). \tag{13}$$

*Accepted for the 38$^{th}$ Conference on Uncertainty in Artificial Intelligence* (UAI 2022).

It then holds that $C^\times = (C_1 \cdot C_2 \cdot \ldots \cdot C_f)^\top$. As a first step of the replacement scheme, consider all $X$ nodes on directed paths from $I$ to $\mathrm{PA}(Y)$, that is $W := \mathrm{AN}(\mathrm{PA}(Y)) \cap \mathrm{DE}(I)$. Among these nodes we consider a causal ordering on the induced graph, that is, we choose $i_1, \ldots, i_f$ such that for all $k, \ell \in \{1, \ldots, f\}$ with $k < \ell$, we have $X^{i_\ell} \in \mathrm{ND}_{\mathcal{G}_W}(X^{i_k})$ (ND denotes the "non-descendants"), where $\mathcal{G}_W$ is the subgraph of $\mathcal{G}$ over nodes in $W$. We now start from the equation $X^{\mathrm{PA}(Y)} = X^{\mathrm{PA}(Y)}$ and replace, on the right-hand side, $X^{i_1}$ by its structural equation, yielding

$$X^{\mathrm{PA}(Y)} = C_1 \cdot X^{\mathrm{PA}_1} + h_1(H, W^c, \varepsilon^{X_{i_1}}),$$

where $\mathrm{PA}_1 = \mathrm{PA}(Y) \setminus \{X^{i_1}\} \cup \mathrm{PA}_{\mathcal{G}_W}(X^{i_1})$ and the $h_1$ term collects error terms and variables not in $W$. $C_1$ is a matrix with dimension $|\mathrm{PA}(Y)| \times |\mathrm{PA}_1|$. We did not replace the variables in $\mathrm{PA}(Y) \setminus \{X^{i_1}\}$, the corresponding submatrix in $C_1$ is the identity. All directed paths from $I$ to $\mathrm{PA}(Y)$ go through $\mathrm{PA}_1$. Condition (B1) therefore implies $|\mathrm{PA}_1| \geq |\mathrm{PA}_1(Y)|$. The row corresponding to $X^{i_1}$ contains the path coefficients from $\mathrm{PA}(X^{i_1})$ to $X^{i_1}$, which are $\mu$-almost surely non-zero. Thus, $C_1$ has $\mu$-almost surely rank $|\mathrm{PA}(Y)|$. We now repeatedly (for $k \in \{2, \ldots, \ell\}$) substitute the variable $X^{i_k}$ in $X^{\mathrm{PA}_{k-1}}$ with its structural equation yielding

$$X^{\mathrm{PA}(Y)} = C_1 \cdot C_2 \cdot \ldots \cdot C_k \cdot X^{\mathrm{PA}_k} + h_k(H, W^c, \varepsilon^{X_{i_1}}),$$

where $\mathrm{PA}_k = \mathrm{PA}_{k-1} \setminus \{X^{i_k}\} \cup \mathrm{PA}_{\mathcal{G}_W}(X^{i_k})$ and $C_k$ contains an identity matrix for the submatrix, corresponding to $\mathrm{PA}_k \cap \mathrm{PA}_{k-1}$ and in the row corresponding to $X^{i_k}$ a vector of coefficients. With the same arguments as above, we have that $|\mathrm{PA}_k| \geq |\mathrm{PA}(Y)|$. (Indeed, otherwise, all directed path would go through a set of nodes of size strictly smaller than $|\mathrm{PA}(Y)|$.) Furthermore, $C_k$ has rank at least $|\mathrm{PA}(Y)|$. (Indeed, if $|\mathrm{PA}_{k-1}| > |\mathrm{PA}(Y)|$, then $C_k$ contains a $|\mathrm{PA}(Y)| \times |\mathrm{PA}(Y)|$ submatrix that is equal to the identity; if $|\mathrm{PA}_{k-1}| = |\mathrm{PA}(Y)|$, then $C_k$ contains a $(|\mathrm{PA}(Y)| - 1) \times (|\mathrm{PA}(Y)| - 1)$ submatrix that is equal to the identity, $\mathrm{PA}_k \setminus \mathrm{PA}_{k-1} \neq \emptyset$, and the entry corresponding to one of the new parents will be non-zero $\mu$-almost surely.) As $W^c$ can be written as a function of $\varepsilon^X$, the above replacement scheme yields the desired form (13). Since $C^\times = (C_1 \cdot C_2 \cdot \ldots \cdot C_f)^\top$ and all non-zero entries are independent realizations from $\mu$, this proves that $C^\times$ is $\mu$-almost surely of rank at least $|\mathrm{PA}(Y)|$, that is, (A1) holds $\mu$-almost surely.

Regarding (A2): Proposition 9 shows that (A2) holds $\mu$-almost surely.

Regarding (A3): Consider a set $S \subseteq \{1, \ldots, d\}$ with $|S| = |\mathrm{PA}(Y)|$ and $S \neq \mathrm{PA}(Y)$. First, we argue that (B3) (i) implies (A3). To see this, assume $\mathrm{AN}_I[S] \neq \mathrm{AN}_I[\mathrm{PA}(Y)]$. Without loss of generality assume that there is an $i^*$ such that $i^* \in \mathrm{AN}_I[S] \setminus \mathrm{AN}_I[\mathrm{PA}(Y)]$. This implies that the $i^*$th row of $C_{\mathrm{PA}(Y)}$ is entirely zero. Moreover, there is a node $X^j \in S$ such that $i^* \in \mathrm{AN}_I[\{j\}]$, and therefore the entry of

the $i^*$th row of $C_S$ that corresponds to $X^j$ must be non-zero $\mu$-almost surely ($C_{i,j}$ corresponds to the $i$-th component of the total causal effect from $I$ on $X^j$ in the SCM given in (1)). It therefore follows that $\mu$-almost surely it holds that

$$\mathrm{Im}(C_S) \neq \mathrm{Im}(C_{\mathrm{PA}(Y)}). \tag{14}$$

Now consider a set $S$ and assume that (B3) (i) does not hold but (B3) (ii) holds. To argue that (A3) holds, we proceed similarly as in the part of the proof showing that $(B1)$ implies $(A1)$. We consider the graph $\mathcal{G}$ over the nodes $W_{S \cup \mathrm{PA}(Y)} := \mathrm{AN}(\mathrm{PA}(Y) \cup S) \cap \mathrm{DE}(I)$. As before, we construct a causal order and substitute the nodes one after each other. This time, we obtain the equation

$$X^{S \cup \mathrm{PA}(Y)} = C_1 \cdot C_2 \cdot \ldots \cdot C_{f'} \cdot I + h^\times(H, \varepsilon^X)$$

and $C_{\cdot, S \cup \mathrm{PA}(Y)} = (C_1 \cdot C_2 \cdot \ldots \cdot C_{f'})^\top$. With the same argument as above, we conclude that $\mu$-almost surely, the rank of $C_{\cdot, S \cup \mathrm{PA}(Y)}$ is strictly larger than $|\mathrm{PA}(Y)|$. This implies that $\mathrm{Im}(C_S) \neq \mathrm{Im}(C_{\mathrm{PA}(Y)})$ $\mu$-almost surely. (Indeed, if $\mathrm{Im}(C_S) = \mathrm{Im}(C_{\mathrm{PA}(Y)})$, then each column of $C_S$ can be written as a linear combination of the columns of $C_{\mathrm{PA}(Y)}$, which implies that $C_{S \cup \mathrm{PA}(Y)}$ is of rank at most $|\mathrm{PA}(Y)|$.) This completes the proof of Theorem 5. $\quad\square$

## 5 ALGORITHM AND CONSISTENCY

The theoretical identifiability results from the previous sections highlight that the causal coefficient $\beta^*$ can be identifiable even in cases that are considered non-identifiable in classical IV literature. We now propose an estimation procedure called `spaceIV` (**spa**rse **c**ausal **e**ffect **IV**) that allows us to infer $\beta^*$ from a finite data set $(X, I, Y) \in \mathbb{R}^{n \times d} \times \mathbb{R}^{n \times m} \times \mathbb{R}^n$. The procedure is based on the optimization problem $\min_{\beta \in \mathcal{B}} \|\beta\|_0$. It iterates over the sparsity level $s$ and searches over all subsets $S \subseteq \{1, \ldots, S\}$ of predictors for that sparsity level to check whether there is a $\beta \in \mathbb{R}^d$ with $\mathrm{supp}(\beta) = S$ that solves (3). We motivate our estimator by considering a hypothesis test. To obtain finite sample guarantees for the test, we assume that the error term is normally distributed (to obtain asymptotic results [Anderson and Rubin, 1950], such assumptions can be relaxed).

Let us consider a fixed sparsity level $s \in \{1, \ldots, d\}$ and the null hypothesis

$$H_0(s): \quad \exists \beta \in \mathbb{R}^d \text{ with } \|\beta\|_0 = s \text{ such that } \beta \in \mathcal{B}.$$

This hypothesis can be tested using the Anderson-Rubin test [Anderson and Rubin, 1949]. Let $P_I := I(I^\top I)^{-1} I^\top$, then the Anderson-Rubin test statistic is defined as

$$T(\beta) := \frac{(Y - X\beta)^\top P_I (Y - X\beta)}{(Y - X\beta)^\top (\mathrm{Id} - P_I)(Y - X\beta)} \frac{n - m}{m}, \tag{15}$$

*Accepted for the 38$^{th}$ Conference on Uncertainty in Artificial Intelligence* (UAI 2022).

and satisfies $T(\beta) \sim F(n - m, m)$ for all $\beta \in \mathcal{B}$, see also Jakobsen and Peters [2022]. It is known [e.g., Dhrymes, 2012] that the limited maximum likelihood estimator (LIML) minimizes this test statistic. For any set $S \subseteq \{1, \ldots, d\}$, denote by $\hat{\beta}_{\text{LIML}}(S) \in \mathbb{R}^d$ the LIML estimator based on the subset of predictors $X^S$ (adding zeros in the other coordinates). It then holds for all $\beta \in \mathbb{R}^d$ with $\text{supp}(\beta) = S$ that

$$T(\hat{\beta}_{\text{LIML}}(S)) \leq T(\beta). \tag{16}$$

Next, for each sparsity level $s \in \{1, \ldots, d\}$ define

$$\hat{\beta}(s) := \hat{\beta}_{\text{LIML}}\left(\underset{S \subseteq \{1,\ldots,d\}:|S|=s}{\arg\min} T(\hat{\beta}_{\text{LIML}}(S))\right), \tag{17}$$

which can be computed by iterating over all subsets with sparsity level $s$. Then, by (16). the hypothesis test $\phi_s : \mathbb{R}^{n \times d} \times \mathbb{R}^{n \times m} \times \mathbb{R}^n \to \{0, 1\}$ defined by

$$\phi_s(X, I, Y) = \mathbb{1}(T(\hat{\beta}(s)) > F_{n-m,m}^{-1}(1 - \alpha))$$

has valid level for the null hypothesis $H_0(s)$ if the error variables are Gaussian (otherwise it has point-wise asymptotic level).

Motivated by this test, we now define our estimator spaceIV. It iterates over $s \in \{1, \ldots, s_{\max}\}$ and in each step computes $\hat{\beta}(s)$ by exhaustively searching over all subsets of size $s$. Then, either $\phi_s$ is accepted and spaceIV returns $\hat{\beta}_{\leq s_{\max}} := \hat{\beta}(s)$ as its final estimator or it continues with $s + 1$. If none of the tests are accepted, the procedure outputs $\hat{\beta}_{\leq s_{\max}} := \hat{\beta}(s_{\max})$ and a warning indicating that the model assumptions may be violated. The detailed procedure is presented in Algorithm 1.

The proposed spaceIV estimator $\hat{\beta}_{\leq s_{\max}}$ satisfies the following guarantees.

**Theorem 6.** *Consider i.i.d. data from a data generating process of the form* (1) *such that $g(H, \varepsilon^Y)$ is Gaussian and assume (A1) and (A2). Let $s_{\max} \in \mathbb{N}$ be such that $s_{\max} \geq \|\beta^*\|_0$. Then, the following two statements hold. (i) We have*

$$\lim_{n \to \infty} P(\|\hat{\beta}_{\leq s_{\max}}\|_0 = \|\beta^*\|_0) = 1 - \alpha.$$

*(ii) If, in addition, (A3) holds, we have, for all $\varepsilon > 0$ that*

$$\lim_{n \to \infty} P(\|\hat{\beta}_{\leq s_{\max}} - \beta^*\|_2 < \varepsilon) = 1 - \alpha.$$

The proof can be found in Appendix C.

## 5.1 CAUSAL SUBSET IDENTIFIABILITY

It is possible to identify a subset of the causal parents under even weaker conditions. This can be done in an idea similar

---

**Algorithm 1:** `spaceIV`

**Input:** predictors $X \in \mathbb{R}^{n \times d}$, response $Y \in \mathbb{R}^n$, instruments $I \in \mathbb{R}^{n \times m}$, sparsity threshold $s_{\max} \in \{1, \ldots, d\}$, significance level $\alpha \in (0, 1)$

1 Initialize sparsity $s \leftarrow 0$
2 Initialize test as rejected $\phi \leftarrow 1$
3 **while** $s < s_{\max}$ *and* $\phi = 1$ **do**
4      Update sparsity $s \leftarrow s + 1$
5      Set $\mathbf{S}_s$ to be all subsets in $\{1, \ldots, d\}$ of size $s$
6      **for** $S \in \mathbf{S}_s$ **do**
7          Compute LIML-estimator $\hat{\beta}_{\text{LIML}}(S)$
8          Compute test statistic $T(\hat{\beta}_{\text{LIML}}(S))$ in (15)
9      **end**
10      Select $S_{\min} \leftarrow \arg\min_{S \in \mathbf{S}_s} T(\hat{\beta}_{\text{LIML}}(S))$
11      Set $\hat{\beta}(s) \leftarrow \hat{\beta}_{\text{LIML}}(S_{\min})$
12      Test whether $H_0(s)$ can be rejected:
         $\phi \leftarrow \mathbb{1}(T(\hat{\beta}(s)) > F_{n-m,m}^{-1}(1 - \alpha))$
13 **end**
14 Set $\hat{\beta}_{\leq s_{\max}} := \hat{\beta}(s)$

**Output:** Final estimate $\hat{\beta}_{\leq s_{\max}}$ and test result $\phi$

---

to invariant causal prediction [Peters et al., 2016]. Define the hypothesis

$$H_0(S) : \exists \beta \in \mathbb{R}^d \text{ such that } \text{supp}(\beta) = S \text{ and } \beta \in \mathcal{B}$$

and the corresponding Anderson-Rubin test

$$\mathbb{1}(T(\hat{\beta}_{\text{LIML}}(S)) > F_{n-m,m}^{-1}(1 - \alpha)).$$

We then have the following guarantees.

**Proposition 7.** *(i) Consider i.i.d. data of $(I, X, Y)$ from a data generating process of the form*

$$Y := X^\top \beta^* + g(H, \varepsilon^Y),$$

*with $I \perp\!\!\!\perp (H, \varepsilon^Y)$ and $g(H, \varepsilon^Y)$ Gaussian. Then,*

$$\lim_{n \to \infty} P\left(\bigcap_{\substack{S:|S|=|\text{ PA}[Y]| \text{ and} \\ H_0(S) \text{ accepted}}} S \subseteq \text{PA}[Y]\right) \geq 1 - \alpha, \tag{18}$$

*where we define the intersection over an empty index set as the empty set.*

*(ii) Consider now i.i.d. data from a data generating process of the form* (1) *such that $g(H, \varepsilon^Y)$ is Gaussian. If (A1) and (A2) hold, then*

$$\lim_{n \to \infty} P\left(\bigcap_{\substack{S:|S|=M \text{ and} \\ H_0(S) \text{ accepted}}} S \subseteq \text{PA}[Y]\right) \geq 1 - \alpha, \tag{19}$$

*where $M := \min\{|S| : H_0(S) \text{ accepted}\}$.*

*Accepted for the 38th Conference on Uncertainty in Artificial Intelligence* (UAI 2022).

The first statement requires the sparsity $\|\beta^*\|_0$ of $\beta^*$ to be known. It still holds when replacing $|\mathrm{PA}[Y]|$ by any $k$ such that $|\mathrm{PA}[Y]| \le k \le d$. The second statement does not require knowledge of $\|\beta^*\|_0$ and provides a guarantee when increasing the subset size until one has found a set that is accepted. The proof of Proposition 7 can be found in Appendix D.

*Remark* 8 (Allowing for children of $Y$). We now discuss the scenario where some of the covariates are causal descendants of the response $Y$. More precisely, we extend the model in (1) to

$$X := BX + \gamma Y + AI + h(H, \varepsilon^X)$$
$$Y := {\beta^*}^\top X + g(H, \varepsilon^Y),$$

where we assume that the matrix

$$B_{\text{ext}} := \begin{pmatrix} B & \gamma \\ (\beta^*)^\top & 0 \end{pmatrix} \in \mathbb{R}^{(d+1)\times(d+1)}$$

is invertible. Theorem 3 and therefore also the results in Sections 5 and 5.1 still hold when using $C_{\text{ext}} := (A^\top, 0)(\mathrm{Id} - B_{\text{ext}})^{-\top}_{1:d,\cdot}$ instead of $C$. Assumption (A2), however, becomes rather restrictive: If there is a child of $Y$ such that all directed path from $I$ to that child go through $Y$, (A2) is not satisfied as the exact intervention effect on $Y$ is recoverable from that child. In particular, in this generalized setting, Proposition 9 or (B1) and (B2) no longer imply that (A2) holds almost surely.

# 6 NUMERICAL EXPERIMENTS

For the numerical experiments, we consider models of the form (1) with $h(H, \varepsilon^X) = H + \varepsilon^X$, $g(H, \varepsilon^Y) = H + \varepsilon^Y$ and dimensions $d = 20$, $q = 1$ and $m = 10$. We generate 2000 random models of this form using the following procedure:

- Generate a random matrix $B \in \mathbb{R}^{20\times20}$ by sampling a random causal order over $X^1, \ldots, X^{20}$. $B$ then has a zero-structure that corresponds to a fully connected graph with this causal order. Each non-zero entry in $B$ is drawn independently and uniformly from $(-1.5, -0.5) \cup (0.5, 1.5)$. Finally, each row of $B$ is rescaled by the maximal value in each row (using one if it is a zero row).
- Generate a random matrix $A \in \mathbb{R}^{20\times10}$ by sampling each entry independently with distribution Bernoulli$(1/10)$ and setting all diagonal entries to 1.
- Generate the parameter $\beta^* \in \mathbb{R}^{20}$ by sampling two random coordinates uniformly from $\{1, \ldots, d\}$ and setting them to 1. All remaining coordinates are set to zero.
- The random variables $I$, $H$, $\varepsilon^X$ and $\varepsilon^Y$ are all drawn as i.i.d. standard normal.

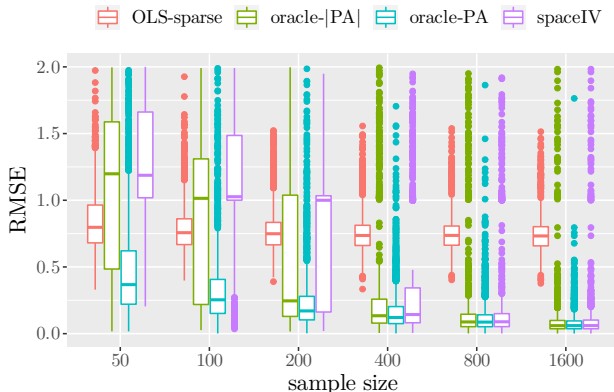

Figure 4: Results for all random models that satisfy (A1)-(A3) (in total 1867 out of 2000 models). The median RSME of the `spaceIV` estimator converges to zero as the simple size increases, which does not hold for `OLS-sparse`. Note that some of the outliers are cut-off in this plot.

For each random model we sample 6 data sets with sample sizes $n \in \{50, 100, 200, 400, 800, 1600\}$. For each data set, we apply the following four methods: (i) `spaceIV`; this is our proposed method described in Algorithm 1 with $s_{\max} = 3$. (ii) `OLS-sparse`; this method goes over all subsets of size at most $s_{\max}$, fits a linear OLS and then selects the subset with the smallest AIC. We also compare our estimator to two oracle methods. (iii) `oracle-|PA|`; this method iterates over all subsets with size 2 (correct parent size), fits the moment equation (3) and selects the best subset in terms of a squared loss based on the moment equation. (iv) `oracle-PA`; this method considers the correct parent set and fits the moment equation (3). Each method results in a sparse estimate $\hat{\beta}$ of $\beta^*$ based on which we compute the root mean squared error (RMSE) given by $\|\beta^* - \hat{\beta}\|_2$.

For each random model, we explicitly check whether the assumptions (A1) and (A3) are satisfied by computing $C$ and verifying the conditions[6]. The results, considering only the random models for which assumptions (A1)–(A3) are satisfied, are given in Figure 4. As expected, `spaceIV` indeed seems to consistently estimate the causal parameter $\beta^*$, while `OLS-sparse` does not. Furthermore, `spaceIV` performs worse as the two oracle methods, illustrating that the estimation in `spaceIV` contains three parts: estimating the correct sparsity, estimating the correct parents set and finally estimating the correct parameters. A mistake in any of these three steps may result in substantial RMSE, which explains the outliers in the plot.

To investigate the consistency of estimating the correct sparsity level in more detail, we consider the fraction of times the correct sparsity level was selected by `spaceIV`. The

---

[6]Assumption (A2) is satisfied by construction because we pick random coefficients for the $B$-matrix, see also (B2).

*Accepted for the 38$^{th}$ Conference on Uncertainty in Artificial Intelligence* (UAI 2022).

result is given in Figure 5. It suggests that the sparsity level is consistently estimated by `spaceIV`.

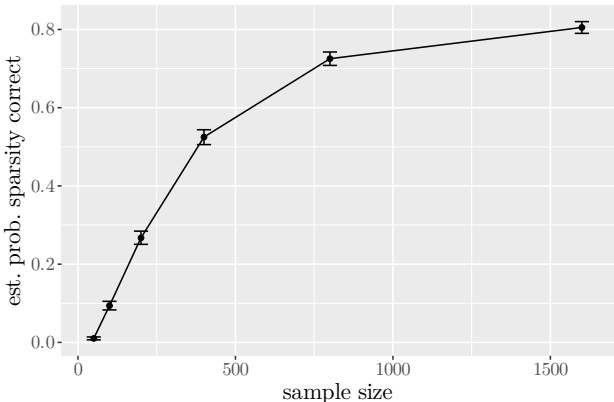

Figure 5: Expected fraction of random models for which `spaceIV` estimated the correct sparsity level. Only random models that satisfy (A1)-(A3) are considered (in total 1867 models). As the sample size increases the estimation of the sparsity level becomes more accurate.

Finally, to investigate the performance of `spaceIV` based on the assumptions (A1)–(A3), we compared the performance of all methods at sample size $n = 1600$ depending on which assumptions are satisfied. (Assumption (A2) is satisfied with probability one, see Proposition 9.) The results are shown in Figure 6. As expected given the theoretical results presented in Section 3.1, `spaceIV` only performs well if all assumptions are satisfied. If only assumption (A1) is satisfied, there are multiple sets with sparsity 2 for which the moment equation (3) can be satisfied. Therefore, while the oracle with the correct parent sets is able to estimate the causal parameter, `spaceIV` and the oracle that only uses the sparsity level may select wrong sets leading to a larger error. Moreover, if none of the assumptions are satisfied the causal parameter is not even identifiable if the true parent set is known.

# 7 CONCLUSION AND FUTURE WORK

We have analysed some of the benefits that come with assuming a sparse causal effect in linear IV models. We have proved identifiability results that make the causal effect identifiable even if there are much less instrument nodes than predictors. Graphical criteria provide intuition on these results and characterize for which graphs the identifiability holds (when randomly choosing coefficients). We have proposed the estimator `spaceIV` and evaluated it on finite samples. The results support our theoretical findings and show that the estimator is often able to find the correct sparsity and the correct parent set.

We believe that the power result for the Anderson-Rubin test

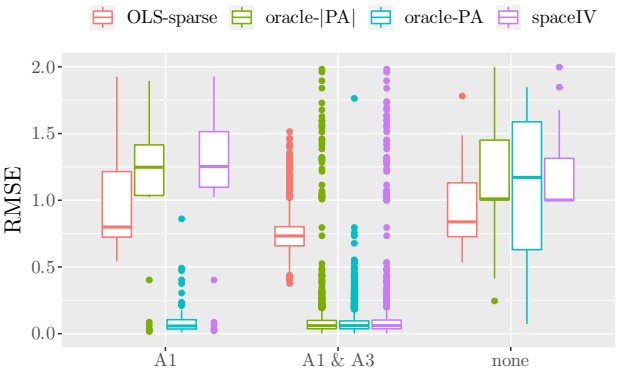

Figure 6: Results for all 2000 random models with $n = 1600$. We split the models into three cases depending on which of the assumptions (A1) and (A3) are satisfied (the group '(A1)' contains 83 models, the group '(A1) & (A3)' contains 1867 models and the group 'none' contains 50 models). If none of the assumptions are satisfied, not even the oracle with known parent set works. If only (A1) is satisfied, multiple sets of size 2 are able to satisfy the moment equation (3) and `spaceIV` may not estimate the correct set. These findings are in par with Theorem 3.

may yield ways for choosing a significance level for finite samples. Furthermore, it could be interesting to investigate to which extent our results generalize to nonlinear models.

## Acknowledgements

NP was supported by a research grant (0069071) from Novo Nordisk Fonden. JP was supported by a research grant (18968) from VILLUM FONDEN.

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
