# OpenReview forum: "Identifiability of Sparse Causal Effects using Instrumental Variables"
_auai.org/UAI/2022/Conference — UAI 2022 Poster_

### Official Review · Reviewer_Z9Vv · 2022-04-05

**Q2(1) Originality/Novelty:** 3
**Q2(2) Significance/Impact:** 3
**Q2(3) Correctness/Technical Quality:** 3
**Q2(6) Clarity Of Writing:** 3
**Q6 Overall Score:** 7
**Q8 Confidence In Your Score:** 4

**Q1 Summary And Contributions:**

The papers considers a causal instrumental variable setting where the number of instruments is small. A new estimator called spaceIV for the causal effect is provided in this context and its performance is evaluated via simulations.

**Q2 Assessment Of The Paper:**

More detailed information regarding each of these aspects is given below:

**Q2(4) Quality Of Experiments (Optional):**

3: Good: The experimental evaluation is adequate, and the results convincingly support the main claims.

**Q2(5) Reproducibility:**

4: Excellent: Key resources (e.g., proofs, code, data) are available and key details (e.g., proof sketches, experimental setup) are comprehensively described for competent researchers to confidently and easily reproduce the main results.

**Q3 Main Strengths:**

The paper is well written, and for the most part, easy to follow.

Assumptions about the models and the corresponding graphs are well motivated and made explicit, with helpful examples and descriptions.

Both graphical criteria and and estimation algorithm are provided, which makes the proposed methodology easy to apply in practice.

**Q4 Main Weakness:**

The paper suffers from some notational issues, mainly, some concepts are only partially defined or not defined at all, making the paper at times harder to read, than it otherwise would have been. (I will list these in more details in the detailed comments).

Section 5 considers the estimation of the causal effect. There seems to be a slight logical jump here, since up until this point, no distributional assumptions have been made (aside from the absolute continuity), but suddenly a test statistic is presented that is supposedly F-distributed. I think the authors should clarify exactly what assumption have to be made about the model so that the test is valid.

The time complexity of Algorithm 1 is not considered. The algorithm involves evaluations over all subsets of specific size, which at a glance would quickly make the proposed estimator unfeasible as the size of the graph increases. In the simulations, the authors consider a graph with 20 X-variables, which seems small to me.

**Q5 Detailed Comments To The Authors:**

There are some notational issues in the paper. In section 2, the authors present the SCM, but it is not fully specified. How are the instruments defined in this model? What are h and g? In section 2.1, the graphical representation is defined somewhat informally. Later, the associated graph G is used, but it has never been defined and operations such as ancestors (AN) are not defined (is a node ancestor of itself for example?)

I do not understand the motivation to call the components of instruments "intervention nodes", since these have nothing to do with interventions defined by the do-operator in the SCM context. I guess this is related to the last paragraph of Section 1, but the motivation is not clear to me.

Algorithm 1 performs possible a large number of hypothesis tests, increasing the likelihood of Type I error. I wonder if the authors have considered to include a correction for this, based on the number of tests made.

In Appendix B, the authors provide a proof of Proposition 7. The claim is that P(AW \in Im(B)) = 0, but at the last step, only an inequality P(AW \in Im(B)) \geq 0 is obtained. Is this a typo? Should the inequality be \leq instead? Also, the final line of the proof mentions "Lemma 7", while this is a proof of Proposition 7.

**Q7 Justification For Your Score:**

I think this is a well written paper, and the results are somewhat novel, but not groundbreaking. I'm concerned about the practical applicability of Algorithm 1, as it seems to only be feasible for small graphs (even if it may scale in the sample size). The related distributional assumptions (gaussian errors etc.) further restrict the potential domain of application.

**Q9 Complying With Reviewing Instructions:**

1: Yes.

---

### Official Review · Reviewer_usjF · 2022-04-12

**Q2(1) Originality/Novelty:** 2
**Q2(2) Significance/Impact:** 2
**Q2(3) Correctness/Technical Quality:** 1
**Q2(6) Clarity Of Writing:** 1
**Q6 Overall Score:** 4
**Q8 Confidence In Your Score:** 2

**Q1 Summary And Contributions:**

The paper  presents  a method for identifying causal effects in a specific instrumental variable setting, where you have multiple instruments I and causes X of a single outcome Y, and the effect of the causes on the outcome is sparse.  The paper presents theoretical results on the identifiability of causal effect vector \beta in this case, a set of graphical criteria for the identifiability,  and a method for learning \beta and the sparsity level s.

**Q2 Assessment Of The Paper:**

More detailed information regarding each of these aspects is given below:

**Q2(4) Quality Of Experiments (Optional):**

2: Fair: The experimental evaluation is weak: important baselines are missing, or the results do not adequately support the main claims.

**Q2(5) Reproducibility:**

1: Poor: Key details (e.g., proof sketches, experimental setup) are incomplete/unclear, or key resources (e.g., proofs, code, data) are unavailable.

**Q3 Main Strengths:**

+ The paper presents some potentially novel and useful ideas
+ while I am not aware of many cases where multiple instruments are available, the authors motivate the work in the setting of multiple experiments which may be represented as multiple instruments.

**Q4 Main Weakness:**

While some of the ideas seem to have merit, the paper seems to be written very hastily to be published as is.
I found the notation very confusing: For example,  in Section 2 instruments and covariates are indexed I_j, X_j, in Section 2.1 they are indexed I^j, X^j and as far as I could tell, in Section for they are just denoted using integers (e.g., S is a set of nodes but takes values in {1, \dots, d}?
These inconsistencies make it very hard to read the paper and understand the technical part, and many notations are not explained very well (for example, what does supp(\beta) in proof of theorem 3 mean? Isn't \beta an element of \matchal \Beta?
In addition, the results are not explained in a very intuitive manner, e.g., why is absolute continuity with respect to Lebesgue measure important for identifiability? How is condition B2 in Section 4 a graphical condition?



**Q5 Detailed Comments To The Authors:**

I suggest that the authors rewrite the paper with a preliminary section where they can be explain all of the notations and definitions used in the text.

**Q7 Justification For Your Score:**

I cannot be confident in any of the results of the paper in its current form, and I believe that the paper should be written more clearly before published.

**Q9 Complying With Reviewing Instructions:**

1: Yes.

---

### Official Review · Reviewer_ts17 · 2022-04-13

**Q2(1) Originality/Novelty:** 2
**Q2(2) Significance/Impact:** 2
**Q2(3) Correctness/Technical Quality:** 2
**Q2(6) Clarity Of Writing:** 2
**Q6 Overall Score:** 4
**Q8 Confidence In Your Score:** 4

**Q1 Summary And Contributions:**

This paper proposes some assumptions under which the causal effect is identifiable even if there are unmeasured confounders and the number of instruments is smaller than the number of observed covariates. The paper also proposes an estimator called spaceIV which can estimate the causal effect if the model is identifiable.

**Q10 Ethical Concerns (Optional):**

There is no ethical concern.

**Q2 Assessment Of The Paper:**

More detailed information regarding each of these aspects is given below:

**Q2(4) Quality Of Experiments (Optional):**

2: Fair: The experimental evaluation is weak: important baselines are missing, or the results do not adequately support the main claims.

**Q2(5) Reproducibility:**

3: Good: Key resources (e.g., proofs, code, data) are available and key details (e.g., proofs, experimental setup) are sufficiently well-described for competent researchers to confidently reproduce the main results.

**Q3 Main Strengths:**

1. The idea of the identifiability assumptions is new.

2. The  introduction of the algorithm for the proposed estimator is clear.

**Q4 Main Weakness:**

1. The proposed assumptions are not  intuitive.

2. There are some errors in the proof of theorems.

3. The proposed estimator  in the algorithm is not consistent with the estimator used in the numerical experiments.

4. The proposed method performs bad when the sample size is not too large and there are many outliers in the estimates even if the sample size becomes large.


**Q5 Detailed Comments To The Authors:**

1. I think in the introduction selection, "this is the case if we ... being independent of X and I" should be changed to "this is the case if we ... being independent of  I." Because if H and \varepsilon^Y are independent of X, then we do not need to use the instrumental variable method.

2. Can you explain more about "More precisely, we can choose I=e_K with e_k, k\in\{1,m\}, being ...and K\sim U(\{1,\ldots,m\})''? Based on the definition of I, I belongs to R^m. I feel they are contradictory.

3. In proposition 2, I think it is better to explain the meaning of the notation "dagger" in the main paper. Because without referring to the appendix, the readers may not understand the meaning of this notation.

4. There are some errors in the proof of Theorem 3, the formulas of (6) and (7) are wrong.

5. Can the proposed assumptions be verified in practice?

6. In the algorithm, it proposes to use the limited maximum likelihood estimator (LIML). But in the numerical experiments, it says that "due to computational reasons, in the experiments we use the two-stage least squares estimator instead of LIML. The former estimator minimises the enumerator of (14)".  On one hand, they are not consistent. On the other hand,  it also shows that LIML is not a good choice in this algorithm.  In addition, can you introduce more about the LIML method?  What is the meaning of "The former estimator minimises the enumerator of (14)''?

7. Based on the simulation results in Figure 4, spaceIV estimator performs bad when the sample size is not too large. The range of the estimated values is very wide and there are some outliers, which means this estimator is not stable.

8. There are some typo errors in the paper. For example, in Figure 6, "A1&A3" needs to be changed to "A1&A2&A3". Please check all the typo errors carefully.


**Q7 Justification For Your Score:**

I make this score based on the main strengths,  weaknesses and my understanding of this paper. I think the weaknesses of this paper slightly outweigh its strengths.

**Q9 Complying With Reviewing Instructions:**

1: Yes.

---

### Official Review · Reviewer_67io · 2022-04-13

**Q2(1) Originality/Novelty:** 3
**Q2(2) Significance/Impact:** 3
**Q2(3) Correctness/Technical Quality:** 3
**Q2(6) Clarity Of Writing:** 3
**Q6 Overall Score:** 7
**Q8 Confidence In Your Score:** 2

**Q1 Summary And Contributions:**

The paper extends the instrumental variable model to sparsely connected linear DAG models with exogenous instrumental variables and an endogenous response variable. They propose the spaceIV estimator for the linear coefficients of the direct causes of the response variable and give rank and graphical conditions under which it is identifiable and unique. SpaceIV can be identifiable when there are more covariates than instruments. The effectiveness of the estimator is validates on synthetic data.

**Q2 Assessment Of The Paper:**

More detailed information regarding each of these aspects is given below:

**Q2(4) Quality Of Experiments (Optional):**

3: Good: The experimental evaluation is adequate, and the results convincingly support the main claims.

**Q2(5) Reproducibility:**

3: Good: Key resources (e.g., proofs, code, data) are available and key details (e.g., proofs, experimental setup) are sufficiently well-described for competent researchers to confidently reproduce the main results.

**Q3 Main Strengths:**

Knowledge of the underlying graph is not required to estimate spaceIV.

The results section is fairly comprehensive and investigates violations of the assumptions and compares against an existing approach and two oracle algorithms.

**Q4 Main Weakness:**

The graphical conditions for identifiability seems to be quite restrictive and simulations suggest that the spaceIV estimator can be quite off when assumptions are violated.

**Q5 Detailed Comments To The Authors:**

Consider defining terms that might not be obvious such as Id and Im.

I recommend providing a discussion about when we might expect the identifiability conditions of this model to hold in the real world.

**Q7 Justification For Your Score:**

The paper provides an interesting extension to the linear instrumental variable framework.

**Q9 Complying With Reviewing Instructions:**

1: Yes.

---

### Decision · Program_Chairs · 2022-05-15

**Decision:**

Accept (Poster)

**Comment:**

Meta Review: I thought the paper provides a novel family of assumptions concerning IV estimation with a large and structured treatment "variable". Results are of much relevance to the UAI community. However, some of the technical parts of the paper need to be cleared from (minor, but distracting) mistakes. The graphical characterization of assumptions does a solid job, but it is still somewhat evolved.